# Overview of Neutralization Assays and International Standard for Detecting SARS-CoV-2 Neutralizing Antibody

**DOI:** 10.3390/v14071560

**Published:** 2022-07-18

**Authors:** Kuan-Ting Liu, Yi-Ju Han, Guan-Hong Wu, Kuan-Ying A. Huang, Peng-Nien Huang

**Affiliations:** 1Research Center for Emerging Viral Infections, College of Medicine, Chang Gung University, Taoyuan 33302, Taiwan; jeff31602@gmail.com (K.-T.L.); lar951185@gmail.com (Y.-J.H.); jacobwu0110@gmail.com (G.-H.W.); kuanyinghuang@gmail.com (K.-Y.A.H.); 2Graduate Institute of Biomedical Science, College of Medicine, Chang Gung University, Taoyuan 33302, Taiwan; 3Division of Infectious Diseases, Department of Pediatrics, Linkou Chang Gung Memorial Hospital, Taoyuan 333, Taiwan; 4International Master Degree Program for Molecular Medicine in Emerging Viral Infections, Chang Gung University, Taoyuan 33302, Taiwan

**Keywords:** neutralizing antibody detection, serological assays, international standard for SARS-CoV-2

## Abstract

We aimed to review the existing literature on the different types of neutralization assays and international standards for severe acute respiratory syndrome coronavirus 2 (SARS-CoV-2). We comprehensively summarized the serological assays for detecting neutralizing antibodies against SARS-CoV-2 and demonstrated the importance of an international standard for calibrating the measurement of neutralizing antibodies. Following the coronavirus disease outbreak in December 2019, there was an urgent demand to detect neutralizing antibodies in patients or vaccinated people to monitor disease outcomes and determine vaccine efficacy. Therefore, many approaches were developed to detect neutralizing antibodies against SARS-CoV-2, such as microneutralization assay, SARS-CoV-2 pseudotype virus assay, enzyme-linked immunosorbent assay (ELISA), and rapid lateral flow assay. Given the many types of serological assays for quantifying the neutralizing antibody titer, the comparison of different assay results is a challenge. In 2020, the World Health Organization proposed the first international standard as a common unit to define neutralizing antibody titer and antibody responses against SARS-CoV-2. These standards are useful for comparing the results of different assays and laboratories.

## 1. Introduction

Severe acute respiratory syndrome coronavirus 2 (SARS-CoV-2) originated in Wuhan, China, in December 2019 [1,2]. This virus belongs to the Nidovirales order of the Coronaviridae family, which includes SARS-CoV and MERS-CoV, which caused outbreaks in 2003 and 2012, respectively. SARS-CoV-2 causes coronavirus disease (COVID-19) that can manifest as a severe acute respiratory syndrome. In March 2020, the World Health Organization (WHO) declared COVID-19 a global pandemic. By April 2022, the total disease burden of COVID-19 was more than 500 million people, and over six million fatalities had been recorded worldwide.

The symptoms of COVID-19 include fever, dry cough, and tiredness, as well as more severe symptoms, such as acute respiratory distress syndrome (ARDS), coagulation disorders, multiorgan dysfunction, and central nervous system infection. The gold standard for diagnosing SARS-CoV-2 is the real-time reverse transcription polymerase chain reaction (qRT-PCR), which involves the detection of viral RNA [3]. However, in patients with an asymptomatic and mild infection, a low PCR positivity rate has been reported in samples collected 8 days after onset of symptoms. A recent study found an increase in the number of asymptomatic cases of COVID-19, suggesting an unmet need for the serodiagnosis of COVID-19. The currently available enzyme immunoassays for detecting exposure to SARS-CoV-2 or vaccine immunization are based on the detection of serum immunoglobulin (Ig) A, IgM, IgG, or total antibodies against the virus [4,5,6].

SARS-CoV-2 has four structural proteins—spike (S), nucleocapsid (N), envelope (E), and membrane (M), with S and N viral proteins being the most immunogenic [6]. The N protein facilitates viral replication, assembly, and release, while the S protein mediates the binding of the virus to the angiotensin-converting enzyme 2 (ACE-2) cellular receptors for virus entry [6,7]. The S protein comprises two subunits, S1 and S2, responsible for binding to host cell receptor (ACE-2) and fusion of viral and cellular membranes, respectively [8,9]. Many serological assays for SARS-CoV-2 have S or N proteins as their target antigens.

The serodiagnosis of the SARS-CoV-2 neutralization antibody (NAbs) needs to be explored for an accurate and reliable diagnosis. Several enzyme-linked immunosorbent assays (ELISAs) and other NAbs testing assays, such as chemiluminescence-based immunoassays and lateral flow (rapid diagnostic) assays, are now available from different manufacturers. The detection accuracy of neutralization IgG ELISA may vary considerably among the test kits, highlighting the need for validation before clinical and research use. Here, we have described SARS-CoV-2 NAbs detection assays, which can be used to address different diagnostic requirements. We analyzed their performance in relation to the neutralization assay, which is the gold standard for assessing immunity against SARS-CoV-2.

## 2. Gold Standard for Neutralizing Antibody Detection and High-Throughput Neutralizing Test

The COVID-19 pandemic triggered a global health emergency, and tremendous efforts have been made to control the pandemic, among which NAbs are of specific interest to researchers. NAbs are generated within weeks after immunization or infection and can provide protective immunity. Thus, the production of NAbs is the main goal of SARS-CoV-2 vaccination and NAbs may be used for patient treatment in the form of monoclonal antibodies. Until now, NAbs detection has been important in vaccine development and determining seroprevalence to assist the government in adjusting policy decisions. The plaque reduction neutralization test (PRNT) is considered the gold standard for measuring neutralization antibodies against SARS-CoV-2 [10,11,12]. Briefly, serial dilutions of the patient specimen (sera) are incubated with the target virus to form an immune complex. The immune complex is then incubated with a cell monolayer and covered with agar to prevent virus diffusion. The last step is to visualize the plaques, which constitute cytopathic effects on the monolayer cells. However, this method is time-consuming and is limited to 6- or 24-well plate format. Thus, it is not suitable for emergency responses, as well as large-scale serology studies.

To accelerate the detection rates, the focus reduction neutralization test (FRNT) was developed and utilizes a specific antigen antibody conjugated with horseradish peroxidase (HRP) to visualize the foci of infected cells. Compared to PRNT, FRNT has a shorter process and can be completed in 3 days. In addition, it uses a 96-well plate, which allows the processing of more samples, and can be read out using the Enzyme-linked immunospot (ELISPOT) reader [13]. Besides the FRNT, the reporter virus is another strategy to detect neutralization titer by visualizing the foci. Due to the characteristics of the reporter virus, no antibodies are used, and no time is spent on blocking, antibody probing, and washing. Thus, the diagnostic procedure is rapid and takes approximately 24 h. Results are read out using the ELISPOT reader [14]. Above, the microneutralization assay is an alternative high-throughput method based on PRNT; it uses the 96-well plate and requires less sample volume to quantify neutralizing antibodies [15,16,17]. Despite several limitations of PRNT, including the use of more reagents and a large sample volume, it is a high-sensitivity assay for detecting neutralization antibodies against live viruses.

## 3. Application of Pseudovirus in SARS-CoV-2 Neutralization Assay

SARS-CoV-2 is a high-risk transmission pathogen, which means that serologic investigations of neutralization antibodies that require handling of live viruses can only be performed in biosafety level 3 facilities. This limitation can delay the development of new insights into vaccine or drug development, especially in a pandemic situation that requires an urgent response. A pseudovirus is a chimeric virus comprising a core skeleton surrounded by a surface protein of the virus of interest. The internal genes of the pseudovirus are modified to hinder them from synthesizing their own surface proteins and thus deter second-round replication [18,19]. The pseudovirus can be handled in biosafety level 2 laboratories, which are available in many facilities. Various SARS-CoV-2 variants have emerged and scientists can use molecular technology to generate point mutations in the pseudovirus consistent with the dominant circulating strain [20]. For high throughput purposes, the pseudovirus harbors the reported gene (NanoLuc luciferase or green fluorescent genes) for the detection of viral-infected cells. Retroviruses are candidates for the construction of pseudovirus vectors. Vesicular stomatitis virus (VSV) and human immunodeficiency virus type 1 (HIV-1) are the commonly used viruses in the SARS-CoV-2 pseudovirus packaging system, based on their high delivery efficiency and allowing insertion of 8 to 9 kb of the interested genes [21,22]. It has been reported that increased S protein expression or increased infectious particle production in the VSV system can be achieved by deleting 18 or 19 amino acids of the cytoplasmic tail and modifying the amino acids at R282Q or D614G [23,24,25]. The VSV harboring the S protein has been applied to pseudovirus neutralization assay for quantifying SARS-CoV-2 neutralization antibodies. In addition, many cell lines have been tested in the VSV-based pseudovirus neutralization assay, including human ACE2 293T (renal epithelia), human Calu-3 (lung epithelial), Vero-E6 (renal epithelial), Ips-CMs (iPS-derived cardiomyocytes), and Huh-7 (human hepatoma) cells, and have shown substantial efficiency [26,27]. Of note, VSV systems that truncate spike proteins in the cytoplasmic tail of SARS-CoV-2 can obtain higher antibody titer in the HIV-1 containing a *Gaussia* luciferase (HIV-Gluc) pseudovirus system. The HIV-Gluc systems harboring the spike protein modification of D614G and R682Q could increase the production of infectious particles [24]. Except for a mono reporter system, a dual reporter system that combines NanoLucluciferase and GFP reporter genes encoded in the HIV-1 pseudovirus can be quantitatively measured through microscopy or flow cytometry. The dual reporter system ensures flexibility in measuring neutralization antibodies [28]. Although the pseudovirus system facilitates S protein studies, some concerns remain. In the pseudovirus-based experiment, cell lines express exogenous human ACE2 only or co-express transmembrane Serine Protease 2 (TMPRSS2) to facilitate the pseudovirus infectivity [29,30]. The S protein density in the viral surface may influence cell entry and neutralization ability [31,32]. The accuracy of the pseudovirus system in neutralization assays still relies on the plaque reduction neutralization test for comparison with the live virus. Before establishing a pseudovirus neutralization assay, cell line susceptibility, optimal cell number, and viral dose range should be carefully investigated.

## 4. Detect Neutralizing Antibodies Using Surrogate Assays

Besides the viral neutralization test (VNT), a non-viral neutralizing antibody detection platform can be utilized. Using the surrogate viral neutralization test (sVNT), biosafety level 1 or 2 (BSL-1 or BSL-2) facilities are adequate for sample processing and antibody detection. Therefore, many immunoassays are used for serum neutralizing activity. The application of surrogate assays not only predicted humoral protection, but also vaccine efficacy during clinical trials and after large-scale vaccination [33]. ELISA, which is based on antigen-antibody reaction, has been applied to detect many biological molecules and as a diagnostic tool [34]. Since the COVID-19 outbreak, many ELISAs have been developed to detect SARS-CoV-2 antibodies, especially neutralizing antibodies. ELISAs are cost-effective and less time-consuming compared to the pseudovirus neutralization test. Based on the interaction between ACE2 and receptor binding domain (RBD), ELISAs usually use recombinant RBD protein or ACE2 as the coating antigen, and HRP-labeled ACE2 or HRP-labeled RBD is used to produce the detection signal [35]. Because the most potent neutralizing antibodies recognize the spike protein RBD, followed by the S1 domain, the spike protein trimer, and the S2 subunit [36], most ELISAs utilize RBD as a target antigen. In addition, four major types of ELISAs, including direct, indirect, sandwich, and competitive, are used. Competitive ELISAs are commonly used for detecting SARS-CoV-2 neutralizing antibodies. The principle of the assay entails the competitive inhibition enzyme immunoassay technique, in which the neutralizing antibodies in serum compete with the HRP-ACE2 or coated ACE2 to bind to the coated RBD or HRP-RBD, respectively. If the serum sample contains SARS-CoV-2 neutralizing antibodies, the intensity of the color produced is inversely proportional to the amount of SARS-CoV-2 neutralizing antibodies [33,37] (Figure 1).

It is also possible to detect the specific type of neutralizing antibody, including IgG, IgA, and IgM, using indirect ELISA [38]. Besides ELISA, chemiluminescence immunoassay (CLIA), chemiluminescence microparticle immunoassay, enzyme-linked fluorescent assay, and fluorescent microsphere immunoassay are also used for neutralizing antibody detection, and related 59 SARS-CoV-2 serology/antibody tests have been granted Emergency Use Authorization (EUA) by the Food and Drug Administration [39]. Most of them are qualitative, but some are semi-quantitative or quantitative [40]. The sensitivity and specificity of ELISA-based serological assay detection are approximately 65% to 98% and 71% to 100%, respectively. CLIA-based serological assay detection shows higher sensitivity and specificity of 77% to 100% and 90% to 100%, respectively, and takes approximately 30 min [41]. However, chemiluminescence equipment is required for CLIA. Therefore, numerous assays of neutralizing antibody detection are available and can be chosen according to need. Neutralizing antibody detection can help determine whether the measured antibody is correlated to the functional antibodies that can keep a population safe during a pandemic [42].

## 5. Lateral Flow Assay of Neutralizing Antibody

Given the time-consuming nature of ELISA/CLIA-based serological assay and the increasing number of COVID-19 cases, the lateral flow assay (LFA) is an ideal alternative for rapid diagnosis. LFA can detect either SARS-CoV-2 antigens or antibodies and provides results within 15–20 min; the results can be observed directly and visually or with a signal detector [43]. Due to its low cost and high availability, the antigen rapid diagnostic test is commonly used for the detection of SARS-CoV-2 protein [44]. At the same time, a serological antibody rapid diagnostic test has been developed for continuous monitoring of the neutralizing antibodies; it can identify individuals with an adaptive immune response to SARS-CoV-2, indicating recent or prior infection [35,43,45]. It is also useful as a necessary adjunct to all available neutralization tests [46]. In addition to diagnosis, LFA can potentially be used in vaccine development by tracing the level of neutralizing antibodies from clinical subjects. As of May 2022, 24 serological antibody rapid diagnostic tests from various manufacturers have been granted emergency use authorization (EUA) by the FDA [39]. The target biomarkers of serological antibody LFA are often COVID-19-specific IgG and/or IgM antibodies. Structurally, each LFA is based on a support card that spontaneously delivers the liquid sample by capillary action. The sample pad is typically at the proximal end of the assay cartridge and is soaked with sufficient sample. The soaked liquid sample enters a second zone, the conjugate pad, which contains a dry conjugate that captures the target analytes in the migrating liquid sample [46]. Neutralizing antibodies in the sample are detected by antigen-gold nanoparticles (Ag-NPs) conjugates within the conjugate pad. The antigens are typically the S and/or RBD, which bind their respective serum antibodies [47]. In the control line region, a signal-transducing colloidal gold nanoparticle (AuNP) antibody binds its capture antibody, while in the test line region, the coated antibody (anti-Human IgG or IgM) captures the AuNP-Ab conjugate-antigen complex (Figure 2). In addition, dedicated instruments, such as fluorescence analysis devices, can be used to provide qualitative or semi-quantitative results.

Recently, a novel fluorescent-based LFA for rapid detection and quantification of total binding antibodies has been reported [48]. Although LFA has the advantages of low cost, high speed, and simple operation, the combined sensitivity of LFAs for COVID-19 was only 66% and showed lower specificity compared to ELISA or CLIA-based detection [49]. Also, LFA was limited to admitted patients only [50] and thus may not be sensitive enough for diagnosing COVID-19 patients in the early stages of illness, as well as in determining vaccine efficacy [43,51]. Nevertheless, LFA can be a preliminary test to hint at potential immune response for further and more robust investigation using surrogate assays.

## 6. Limitations of Surrogate Assays and Rapid Lateral Flow Tests for SARS-CoV-2 Neutralization Antibodies

Surrogate assays for measuring neutralizing antibodies have been indispensable during the COVID-19 pandemic and can be performed at lower biosafety level facilities. A comparison of sVNT and microneutralization assay showed a correlation coefficient (R^2^) above 0.7, indicating a relatively well performance [33,52,53]. Besides, sVNTs are faster and simpler than traditional detection methods. However, the sVNT cannot detect all neutralizing antibodies in the patient sample. As previously demonstrated, neutralizing antibodies bind not only to the RBD but also to the N-terminal of spike proteins or other surface proteins on SARS-CoV-2 [54]. Notably, a previous study did not find a high correlation between the surrogate assay and the neutralization titer, possibly because of some neutralizing antibodies against other surface proteins on SARS-CoV-2 in addition to RBD [52,55]. Thus, surrogate assays can bias the detection of neutralizing antibodies. Further, most reports have used linear regression to analyze surrogate assay results. However, huge variations were discovered in the lower 25th and the higher 75th percentile [55].

A rapid lateral flow test is another approach for detecting SARS-CoV-2 neutralizing antibodies [35,56]. It is based on determining the color intensity in a test line to show the level of the neutralizing antibody. Although it can be highly correlated with the microneutralization titer, it cannot accurately reflect the level of the titer value. Hence, the rapid lateral flow test is not universally used to measure neutralizing antibodies. Despite these limitations of sVNT and rapid lateral flow test, they are still powerful approaches to accelerate vaccine development and satisfy the requirement for mass detection of neutralizing antibodies.

## 7. International Standard for SARS-CoV-2 and Its Importance for Calibrating Measurements in Serological Assays

An international standard, comprising a common unit and value of neutralizing antibody titer and antibody responses, is important for comparable diagnosis. Despite the development of many neutralization antibody tests, it is difficult to compare the results of different assays. Besides, the variation of experimental results in different laboratories may be due to disparity in interpretation, detection equipment, and protocol. Therefore, international standards for SARS-CoV-2 diagnosis are required to calibrate measurements from different methods or laboratories.

The significance of international standards in emerging viral infections has been proposed previously [57,58,59]. During, the outbreak of Zika virus disease in Latin America in 2015–2016 [60], the WHO released the international standard for immunologic assays for Zika virus to accelerate vaccine development and improve the consistency of quantification [61]. In addition, the first international standard for antiserum to Respiratory Syncytial Virus has been reported previously [62].

The first WHO International Standard and International Reference Panel for anti-SARS-CoV-2 immunoglobulin were adopted by the WHO Expert Committee on Biological Standardization in December 2020 [63,64]. The International Standard was based on pooled human plasma from SARS-CoV-2 convalescent patients, and antibodies were tested for use as a reference panel. The reference panel included sample F (20/150), high titer; sample J (20/148), mid titer, sample E (20/144), low anti-S and relatively high anti-N protein antibodies; sample I (20/140) low titer, and sample H (20/142), negative (Table 1).

In addition to the international standard, the binding antibody unit (BAU) is a unit proposed by WHO and represents the amounts of antigen-specific antibodies in a sample [65]. For instance, the antibody responses in the WHO reference panel include anti-S1, anti-spike, anti-RBD, and anti-N, and their levels can be expressed in BAU. Scientists can use BAU in many serological assays, such as microneutralization, pseudotype, and surrogate assays, for detecting antibody responses against SARS-CoV-2. Recently, many studies have used the international standard as a unit to represent the amount of neutralizing antibodies in the sample, and this has enabled comparisons to the reference value [66,67].

During the COVID-19 pandemic, many candidate vaccines underwent clinical trials [68,69]. With the high demand for COVID-19 vaccines worldwide, more licensed vaccines should be availed, particularly in the low- and middle-developed countries. However, a candidate vaccine must complete the clinical trial before it is licensed for public use; this process is time-consuming, and it is sometimes difficult to find eligible clinical trial participants. Therefore, immunological correlates of protection are proposed to replace traditional large-scale clinical trials [70,71,72]. Correlates of protection are based on quantification of neutralizing antibody responses and comparison of neutralizing antibody titers between candidate and listed vaccines. Previous studies reported that the neutralizing antibody level is highly proportional to vaccine efficacy and protection [73], indicating why the correlates of protection are feasible approaches for accelerating the development of vaccines and their certification. Given the suggested importance of the parallel neutralizing level comparison between candidate and listed vaccines, the international standard becomes a powerful reference and enabler of the correlates of protection. Undoubtedly, international standards for SARS-CoV-2 are vital biological references for serological assays and common units for calibrating neutralizing antibody measurements.

## 8. Conclusions

NAbs titer is a key factor for predicting immunity against SARS-CoV-2. Neutralization assays are powerful tools that have been used for COVID-19 diagnosis and vaccine evaluation. Various mature methods for detecting NAbs have been useful in the development of vaccines. However, most of these methods have advantages and shortcomings that need to be improved in the future. For instance, PRNT has safety issues, as it requires exposure to live viruses in the biosafety level 3 laboratory. Also, it remains unclear how the dynamic changes in the NAbs titers are correlated with patient clinical outcomes. Some sVNTs have been developed with high-throughput and time-saving features. The high-throughput assay may become a potential alternative gold standard for detecting and measuring NAbs. Scientists and virologists need to be innovative and develop more convenient methods for NAbs measurement in the future. Additionally, the heterogeneous design of the virus neutralization and binding assays limits the quantitative comparison of the results. These technical hurdles can be alleviated by the incorporation of an international standard.

Moreover, the natural effective level of NAbs required to prevent SARS-CoV-2 infection is unknown. A precise evaluation of the protective immunity effective at both the individual and population levels is difficult. Nevertheless, the NAbs titer is crucial for guiding decision-makers in easing the strict COVID-19 restrictions. An effective and time-effective neutralization assay can monitor in real-time the changes in the protective neutralization titers of SARS-CoV-2-infected patients and vaccinated individuals. Additionally, it can help in screening effective vaccine candidates. With the high mutation rate of SASR-CoV-2, the circulating strains keep changing and make the antigenicity alteration. The antigenicity alteration affects the interaction between RBD and ACE2 where the neutralizing antibodies most frequently bind. Hence, the next generation of neutralization assays should be designed against the mutated viruses in the future. In sum, scientists should make effort to further improve the specificity and sensitivity of novel neutralization assays, which are key in preparing for future pandemics, such as COVID-19.

## Figures and Tables

**Figure 1 viruses-14-01560-f001:**
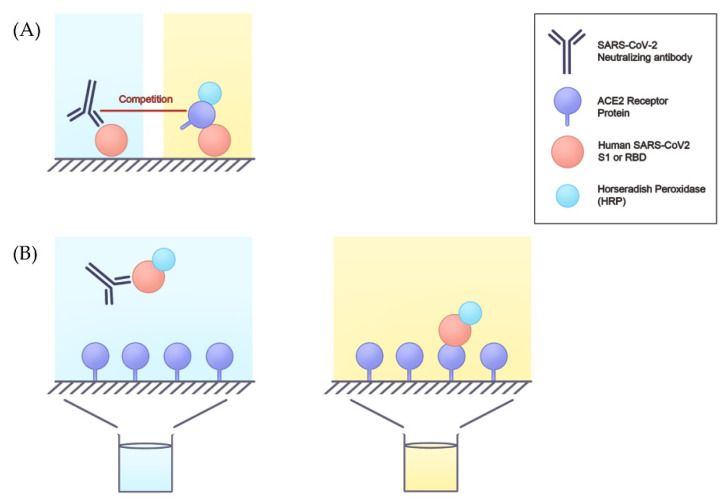
The principle of neutralizing antibody detection using ELISA. Based on the competitive ELISA, one of the methods (**A**), the HRP-ACE2 competes with SARS-CoV-2 neutralizing antibody to bind to the antigen-like S1 or RBD. The light signal is strong when the level of the neutralizing antibody is low. In the other method (**B**), the neutralizing antibody is pre-incubated with antigen-like S1 or RBD and then added to the well that is coated with ACE2. After the wash step, the antigens not recognized by the neutralizing antibodies can bind to the well plate (ACE2), and a bright color can also be detected. ELISA, enzyme-linked immunosorbent assay; HRP-ACE2, horseradish peroxidase-angiotensin-converting enzyme 2; RBD, receptor binding domain; SARS-CoV-2, severe acute respiratory syndrome coronavirus 2; S1, spike protein 1.

**Figure 2 viruses-14-01560-f002:**
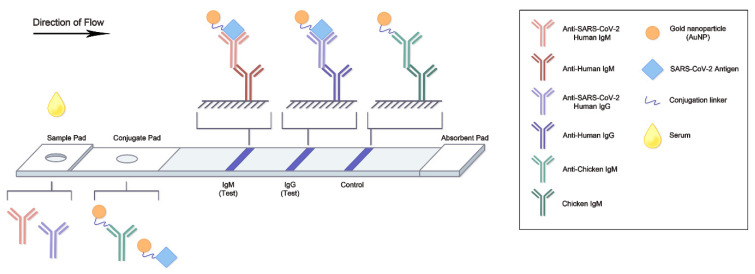
The scheme of neutralizing antibody LFA. After the drop of serum sample is placed on the sample pad, the sample moves to the conjugate pad by capillary action. In this part, the AuNP-antigen can be recognized by the neutralizing antibody. Antibody detection, including IgM and IgG, occurs in the next part; the antibody and AuNP-antigen complex show a signal in the test region. AuNP, gold nanoparticle; Ig, immunoglobulin; LFA, lateral flow assay; SARS-CoV-2, severe acute respiratory syndrome corona virus 2.

**Table 1 viruses-14-01560-t001:** WHO international reference standards for anti-SARS-CoV-2 immunoglobulin and neutralizing antibody.

	WHO Reference Standards						
sample code	20/130	20/136	20/150	20/148	20/144	20/140	20/142
Neutralizing antibody (IU/mL)	1300	1000	1473	210	95	44	-
Anti-RBD IgG (BAU/mL)	502	1000	817	205	66	45	-
Anti-S1 IgG (BAU/mL)	588	1000	766	246	50	46	-
Anti-Spike IgG (BAU/mL)	476	1000	832	241	86	53	-
Anti-N IgG (BAU/mL)	747	1000	713	295	146	12	-

IU: international unit; BAU: binding antibody unit; Ig, immunoglobulin; S1, spike protein subunit 1; RBD, receptor binding domain; N, nucleocapsid; WHO, World Health Organization. -: Negative.

## Data Availability

Not applicable.

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
