# Peer review of "Overview of Neutralization Assays and International Standard for Detecting SARS-CoV-2 Neutralizing Antibody"

_viruses, 2022, doi:10.3390/v14071560_

Round 1

Reviewer 1 Report

Liu et al give a valuable introduction to the SARS-CoV-2 Nab assays and international standards. There are some minor issues that should be addressed before acceptance.

1.     In line 37, please provide the accurate date of the numbers due to the evolving pandemic.

2.     In lines 44-48, could the antibody assays be employed as diagnostic approaches? The asymptomatic cases couldn’t be identified using the PCR assay? When the PCR assay reports negative, could the individual shed the virus or not?

3.     In line 92 “no time is spent in blocking, antibody probing, and washing”, the HRP-conjugated antibody needs no incubation?

4.     What’s the difference between Vero CC81 and the Vero E6? It will be clearer to provide a scheme for the FRNT.

5.     What’s the correlation between PRNT and FRNT?

6.     The 293T cell is usually known as unsusceptible to SARS-CoV-2 infection.  When used as SARS-CoV-2 target cells, hACE2 and/or TMPRSS2 are usually overexpressed in this cell line. Please check that in line 123.

7.    In lines 128-129, The HIV-Gluc systems harboring the spike protein modification of D614G and R682Q could increase production of infectious particles”. Please check and confirm whether these mutations yield enhanced production or the infectivity of the particles.

8.     In line 135, TMPRS2 should be transmembrane protease serine 2 (TMPRSS2).

9.     In line 178, the antibody tests were approved by FDA or other agencies? Please clarify.

Reviewer 2 Report

The article ‘Overview of Neutralization Assays and International Standard for Detecting SARS-CoV-2 Neutralizing Antibody’ by Kuan-Ting Liu and collaborators briefly summarized different testing approaches for NAbs of SARS-CoV-2, along with internationally formulated standard for anti-SARS-CoV-2 immunoglobulin and neutralizing antibodies. As the NAbs test for SARS-CoV-2 is pivotal for development of potential vaccines, and guiding authorities to determine the appropriate and flexible policies for pandemic control. Comparing all current NAbs testing for SARS-CoV-2 will help to develop more precise and efficient approaches for COVID-19 diagnosis and vaccine evaluation. Below are suggestions and comments:

1. The 7th part, ‘Limitations of surrogate assays and rapid lateral flow tests for SARS-CoV-2 neutralization antibodies’ should be after the 5th part, ‘Lateral flow assay of neutralizing antibody’. Then introducing the international standard for SARS-CoV-2 NAbs, and explain why this international standard is so important and necessary? Then how this standard can be incorporated into previously described assays, and what can be improved for those assays for unified data interpretation? Most importantly, this standard will be helpful for future potential vaccine evaluation, and to eliminate the misinterpretation of indicated vaccines by different organizations.

2. The discussion part. As the pandemic is still ongoing, the pace of newly emerged variants is faster and the window period for the future globally circulating variants is much shorter. Is it possible to further address what potential challenges for current NAbs assays to better accommodate the future challenges?

Minor issues:

Line 214: The ‘period’ is missing at the end of the last sentence. 

Line 315: ‘methods for Nabs’ should be changed into ‘methods for NAbs’.
